# Fe_3_O_4_ Hollow Nanosphere-Coated Spherical-Graphite Composites: A High-Rate Capacity and Ultra-Long Cycle Life Anode Material for Lithium Ion Batteries

**DOI:** 10.3390/nano9070996

**Published:** 2019-07-10

**Authors:** Fuyi Jiang, Xinsheng Yan, Rong Du, Litao Kang, Wei Du, Jianchao Sun, Yanli Zhou

**Affiliations:** School of Environmental and Materials Engineering, Yantai University, Yantai 264005, China

**Keywords:** spherical-graphite, Fe_3_O_4_, anode materials, full cell, lithium ion batteries

## Abstract

The spherical-graphite/Fe_3_O_4_ composite has been successfully fabricated by a simple two-step synthesis strategy. The oxygenous functional groups between spherical-graphite and Fe_3_O_4_ benefit the loading of hollow Fe_3_O_4_ nanospheres. All of the composites as anodes for half cells show higher lithium storage capacities and better rate performances in comparison with spherical-graphite. The composite containing 39 wt% of hollow Fe_3_O_4_ nanospheres exhibits a high reversible capacity of 806 mAh g^−1^ up to 200 cycles at 0.5 A g^−1^. When cycled at a higher current density of 2 A g^−1^, a high charge capacity of 510 mAh g^−1^ can be sustained, even after 1000 long cycles. Meanwhile, its electrochemical performance for full cells was investigated. When matching with LiCoO_2_ cathode, its specific capacity can remain at 137 mAh g^−1^ after 100 cycles. The outstanding lithium storage performance of the spherical-graphite/Fe_3_O_4_ composite may depend on the surface modification of high capacity hollow Fe_3_O_4_ nanospheres. This work indicates that the spherical-graphite/Fe_3_O_4_ composite is one kind of prospective anode material in future energy storage fields.

## 1. Introduction

Lithium-ion batteries (LIBs) as the most advanced energy storage systems have been widely used in electric vehicles and some portable electronic appliances due to the good cycle life and high energy density [1,2]. Graphite is the commonly used anode material for LIBs, and natural graphite has gained great attention due to its high conductivity, large abundance, excellent cycling stability, and low cost features. However, its low theoretical specific capacity (372 mAh g^−1^) as an anode material is unable to meet the rising energy requirements of high energy and power densities [3,4,5]. Meanwhile, the voltage plateau of graphite is low—close to the lithium—which results in the growth of lithium dendrites on the graphite surface. The formed lithium dendrites can not only reduce the battery capacity, but also lead to serious safety accidents [6]. There are several strategies to improve the electrochemical performance of graphite: mechanical grinding [7,8], oxidation treatment [9,10], surface coating [11,12,13,14], and so on. As reported by Wu et al., oxidized natural graphite using air as an oxidant could improve the lithium storage performance of graphite [10]. A hybrid with a high capacity anode material could improve the lithium storage performance of graphite. For example, the S-doped graphite modified by CoO nanoparticles delivered a specific capacity of 440 mAh g^−1^ at a low current density of 150 mA g^−1^ after 100 cycles [12]. The Fe_2_O_3_/graphite composite reported by Wang et al. via ball milling method exhibited an initial charge capacity of 535 mAh g^−1^ at 0.1 A g^−1^, and it could remain at 490 mAh g^−1^ after 55 cycles. However, Fe_2_O_3_ was distributed inhomogeneously on the surface of graphite in the composites prepared by the ball milling method, which led to poor high-rate cycling stability [14].

It has been shown that the introduction of oxygenous functional groups (hydroxyl group, carboxyl group, and epoxy group) to the surface of graphite facilitates the combination of metal oxide and graphite [15,16]. Fe_3_O_4_ is considered as the promising anode material due to the high theoretical specific capacity (924 mAh g^−1^), high electronic conductivity, low-cost, and eco-friendly characteristics [17,18]. Dependent on the surface modification of high capacity Fe_3_O_4_, the coating of Fe_3_O_4_ on the surface of oxidized spherical-graphite (abbreviated as SGO) would obtain high performance electrode materials. Hollow nanostructures have been confirmed to effectively suppress the volume changes of active materials upon repeated cycles, avoiding pulverization of the materials [19,20]. 

In this work, hollow Fe_3_O_4_ nanospheres were chosen to modify SGO. The spherical-graphite (SG) was first oxidized by sulfuric acid to obtain SG with oxygen-containing functional groups, and then numerous hollow Fe_3_O_4_ nanospheres were coated on the surface of the SGO by a simple solvothermal reaction, forming the SGO/Fe_3_O_4_ composites. By controlling the loading mass of Fe_3_O_4_, different composites can be obtained. The electrochemical results show that the SGO/Fe_3_O_4_ composites for half cells exhibit splendid lithium storage performance, owing to the combination of the high capacity of Fe_3_O_4_ and high conductivity of graphite. Among the SGO/Fe_3_O_4_ composites, the composite with Fe_3_O_4_ content of 39 wt% shows the best high-rate capacity and ultra-long cycle life. Using optimized SGO/Fe_3_O_4_ composites as anode materials, the SGO/Fe_3_O_4_/LiCoO_2_ full cells also present good electrochemical performance. 

## 2. Materials and Methods

### 2.1. Oxidation Treatment of SG

The SG was from the Qingdao Qingbei Carbon Products Co. (Qingdao, China). The particle size of SG was about 16 μm to 20 μm. In a typical preparation process, 10 g SG and 100 mL concentrated sulfuric acid (H_2_SO_4_, 98 wt%) were added to the flask, and then the mixture was kept for 10 h at 180 °C with stirring in an oil bath. After the above dispersion was cooled to room temperature, it was harvested by vacuum filtration and washed with alcohol (C_2_H_5_OH, 99.7 wt%) and deionized water several times to a pH of about 7. Then, the product was dried at 60 °C to obtain SGO.

### 2.2. Synthesis of SGO/Fe_3_O_4_ Composites

In a typical synthesis, FeCl_3_ was first dissolved in 30 mL ethylene glycol, then 0.55 g Polyethyleneglycol (PEG, M_w_ = 2000), 2 g sodium acetate (NaAc), and 0.2 g SGO were added into the above solution, respectively. After that, the mixture was vigorously stirred at 60 °C for 2 h. Finally, the above suspension was sealed and transferred into a 50 mL Teflon-lined autoclave and held at 200 °C for 15 h. After the reaction was cooled to ambient temperature, the black precipitate was washed with deionized water and ethyl alcohol several times, and then dried under vacuum to obtain SGO/Fe_3_O_4_ composites. During the preparation process, five different amounts of FeCl_3_ (0.05, 0.1, 0.2, 0.3, and 0.4 g) were used, and the obtained composites were labelled as SGO/Fe_3_O_4_-1, SGO/Fe_3_O_4_-2, SGO/Fe_3_O_4_-3, SGO/Fe_3_O_4_-4, and SGO/Fe_3_O_4_-5, respectively. The preparation process of SGO/Fe_3_O_4_ composites is illustrated in Figure 1.

### 2.3. Material Characterization

The structures of samples were conducted by X-ray diffractometer (XRD-7000X, Shimadzu, Kyoto, Japan) using Cu Kα (λ = 0.15406 nm) and Raman spectrometer (Horiba LabRAM HR Evolution, Horiba Jobin Yvon, Paris, France) using a laser (λ = 532 nm). The morphologies were characterized by transmission electron microscope (TEM, JEOL-1400 plus, JEOL, Tokyo, Japan) and field emission scanning electron microscope (FESEM, JSM-7610F, JEOL, Tokyo, Japan). The weight ratios of Fe_3_O_4_ in the composites were measured by thermogravimetric analysis (TG, Sta 449F3, Netzsch, Selb, Germany) with a ramp rate of 10 °C min^−1^ under air atmosphere from ambient temperature to 1000 °C.

### 2.4. Electrochemical Measurement

The electrode was composed of active material (70 wt%), acetylene black (20 wt%), and carboxylmethyl cellulose (CMC, 10 wt%). The average loading weights of electrodes were approximately 1.0 mg cm^−2^. The areal loading weights and thickness of the electrodes for each sample from SGO/Fe_3_O_4_-1 to SGO/Fe_3_O_4_-5 as well as SGO and SG are shown in Appendix A. The CR2032-type coin half cells were finally assembled in an Ar-filled glove box (H_2_O, and O_2_ < 0.1 ppm), by using metallic lithium plate as the counter electrode, and a Celgard 2400 microporous polypropylene membrane as the separator. The electrolyte was 1 M LiPF_6_ in a mixture of dimethyl carbonate and ethylene carbonate (1:1, volume %). The coin cells were measured on a battery testing system (CT2001A, Wuhan, China) in a voltage range of 0.01 to 3 V. Cyclic voltammetry (CV) curves were acquired on a electrochemical workstation (CHI660E, Shanghai Chenhua Instruments, Shanghai, China) between 0.01 to 3 V with a scanning rate of 0.1 mV s^−1^. Electrochemical impedance spectroscopy (EIS) was performed with an electrochemical workstation (PGSTAT302N, Metrohm, Herisau, Switzerland) by using an alternating current (AC) voltage of 10 mV in a frequency range of 100 kHz and 0.01 Hz. For full cells, the anode was made with the above experimental strategy and then it was electrochemically activated for three cycles. The cathode consisted of 80 wt% of LiCoO_2_ (areal loading weights and thicknesses of electrodes are shown in Appendix A), 10 wt% of acetylene black, and 10 wt% of polyvinylidene fluoride (PVDF), with 1-Methyl-2-pyrrolidone (NMP) as dispersant, and then the slurry was spread on metallic aluminum foil. The capacity ratio between anode and cathode was controlled at 1.2: 1. 

## 3. Results and Discussion

X-ray diffraction (XRD) patterns of SG, SGO, and SGO/Fe_3_O_4_-4 composite are shown in Figure 2. As shown in Figure 2a,b, a strong diffraction peak at 26.4° is indexed to (002) the crystal plane of graphite, and no obvious changes of diffraction peaks for SG were observed after oxidation treatment, suggesting that no phase transition occurs in the preparation process of SGO. Figure 2c shows that all the other diffraction peaks are ascribed to Fe_3_O_4_ (JCPDS NO. 19-0629), besides the diffraction peaks of graphite. The diffraction peaks at 30.1°, 35.5°, 43.1°, 56.9°, and 62.5° are attributed to (220), (311), (400), (511), and (440) crystal planes of Fe_3_O_4_, respectively. Similar XRD patterns are also obtained for SGO/Fe_3_O_4_-1, SGO/Fe_3_O_4_-2, SGO/Fe_3_O_4_-3 and SGO/Fe_3_O_4_-5 (Appendix A). Raman spectra of three samples are presented in Appendix A. The intensity ratios of D and G bands (I_D_/I_G_) of SGO (0.22) are lower than that of SG (0.26), which indicates that many defects in SG could be eliminated after being oxidized by sulfuric acid [19,20,21], while the I_D_/I_G_ value of SGO/Fe_3_O_4_-4 increases in comparison with that of SG and SGO, which may be attributed to the formation of many defects and reduction of oxygen-containing groups during the preparation of SGO/Fe_3_O_4_ composites [13,22,23].

The morphologies of SGO/Fe_3_O_4_ composite are displayed in Figure 3. The SEM images show that Fe_3_O_4_ nanospheres are uniformly coated on the surface of SG (Figure 3a,b), and no extra particle agglomeration of Fe_3_O_4_ is observed. Figure 3c presents the scanning electron microscope (SEM) image of SGO/Fe_3_O_4_ composite. It can be found that several Fe_3_O_4_ nanospheres are attached on the surface of SGO, and the average diameter of particles is about 140 nm. The magnified transmission electron microscope (TEM) image shows that Fe_3_O_4_ has a hollow nanostructure (Figure 3d).

Figure 4a shows the CV curves of SGO/Fe_3_O_4_-4 at a scanning rate of 0.1 mV s^−1^. In the first cathodic scan, two peaks at about 0.085 V and 0.17 V are attributed to the Li^+^ insertion into graphite layers. A sharp cathodic peak at approximately 0.68 V corresponds to the electrochemical reduction process of Fe^3+^ and Fe^2+^ to Fe^0^ [24]. In the second cycle, the peak at 0.68 V shifts to 0.82 V owing to the irreversible structural changes [25]. The peak at around 0.85 V is likely attributed to the generation of solid electrolyte interface (SEI) film and electrolyte decomposition [26]. The cathodic peak at about 1.4 V is assigned to Li^+^ insertion into the Fe_3_O_4_. The obvious anodic peak at about 0.21 V is attributed to Li^+^ extraction from the SGO/Fe_3_O_4_-4 electrode [27,28]. The anodic peaks at around 1.55 V and 1.83 V are ascribed to the oxidation of Fe^0^ to Fe^2+^ and Fe^3+^, respectively [24]. Galvanostatic lithiation/delithiation curves of the SGO/Fe_3_O_4_-4 at the first five cycles at 0.1 A g^−1^ are exhibited in Figure 4b. The reversible capacities of SG, SGO, and SGO/Fe_3_O_4_-4 for the first cycle are 411, 426, and 808 mAh g^−1^, corresponding to Coulombic efficiency (CE) of 81%, 82%, and 77% (Appendix A), respectively. The large irreversible capacity was caused by the SEI layer formation due to the electrolyte decomposition and lithium being trapped in the active material, and so on [29,30,31,32]. The formed SEI film is the main factor in the irreversible capacity loss during the first discharge process. Fortunately, for the second cycle, the CE of three samples increased up to 97.1%, 95.2%, and 94.2%. Figure 4c shows the cycling performance of three samples at 0.5 A g^−1^. It can be found that the SGO/Fe_3_O_4_-4 composite exhibits a high reversible capacity of 806 mAh g^−1^ after 200 cycles. The capacity increase may be attributed to the following reasons. The first reason is the self-activation process of Fe_3_O_4_ active materials upon repeated cycles [33,34,35]. The particle size of Fe_3_O_4_ will decrease. The increased specific surface area will form more active sites, and improve the electrolyte accessibility, which can provide more surface-related capacitance [33,34,35]. Second, as the cycle number increases, a more-stable SEI film will form on the surface of the electrode material, which is beneficial for the lithium ion storage [36]. Third, the reversible growth of a polymeric gel-like film resulting from kinetic electrolyte degradation can also lead to the capacity increase [37,38,39]. In addition, SGO (324 mAh g^−1^) presents a slightly higher specific capacity than SG (295 mAh g^−1^). Some structure defects, such as edge carbon atoms, carbon chains, and sp^3^-hybridized carbon atoms, were removed [9], and the functional groups (—CO, —COOH) were formed on the unsmooth surface of the graphite after oxidation [40], which induced the formation of a stable SEI layer and contributed to the good cycling performance. The rate capabilities of the three samples are displayed in Figure 4d. The SGO/Fe_3_O_4_-4 delivers excellent rate capability in comparison with SGO and SG. The average charge capacity cycled at 0.1, 0.2, 0.5, 1, 2, and 5 A g^−1^ is 595, 589, 516, 444, 376, and 299 mAh g^−1^, respectively. The specific capacity can recover to 610 mAh g^−1^ as the current density returns to 0.1 A g^−1^, respectively, indicating good reversibility features of SGO/Fe_3_O_4_-4. The long periodic cycling performance of the three samples at a high current density of 2 A g^−1^ are further discussed (Figure 4e). We found that SGO/Fe_3_O_4_-4 exhibited a high specific capacity of 510 mAh g^−1^ up to 1000 cycles, which is much higher than SG (146 mAh g^−1^) and SGO (114 mAh g^−1^). The outstanding cycling performance of SGO/Fe_3_O_4_ composites could be summarized as follows: (i) the high conductivity of the graphite and hollow nanostructure of Fe_3_O_4_ benefits the electron transfer, and promotes the lithium ion diffusion; (ii) the uniform coating of Fe_3_O_4_ inhibits the formation of lithium dendrites during the charge and discharge process; (iii) the abundant active functional groups on the surface of SG serve as a linker and can facilitate the uniform growth of Fe_3_O_4_, leading to the formation of SGO/Fe_3_O_4_ composites. The effective combination of high capacity Fe_3_O_4_ and SGO contributes to its superior electrochemical performance for LIBs (Appendix A).

The loading amount of Fe_3_O_4_ hollow nanospheres is a key factor that affects the lithium storage performances of SGO/Fe_3_O_4_ composites. SEM images of composites with different loading amounts of Fe_3_O_4_ are shown in Appendix A. More Fe_3_O_4_ nanospheres are adsorbed on the surface of the composite with the increase of FeCl_3_ content. In addition, it can be found that particle agglomeration occurs due to excess of particles (Appendix A). According to TG results, the mass ratios of Fe_3_O_4_ in SGO/Fe_3_O_4_-1, SGO/Fe_3_O_4_-2, SGO/Fe_3_O_4_-3, SGO/Fe_3_O_4_-4, and SGO/Fe_3_O_4_-5 are about 11.6%, 24.7%, 32.5%, 39.2%, and 48%, respectively (Appendix A). Figure 5a displays the cycling performance of five different composites. As observed, SGO/Fe_3_O_4_-4 delivers the highest specific capacity among these composites, which is consistent with those of Figure 5c. The rate performances of different composites are shown in Figure 5b. SGO/Fe_3_O_4_-5 exhibits a slightly high reversible capacity than SGO/Fe_3_O_4_-4 at low current densities due to the increase of Fe_3_O_4_ content. Because the theoretical capacity of Fe_3_O_4_ is higher than that of SGO, a higher capacity can be obtained when the loading amount of Fe_3_O_4_ nanospheres is increased. However, when the loading amount of Fe_3_O_4_ nanospheres is too high, the Fe_3_O_4_ nanospheres will agglomerate together, resulting in the decrease of specific capacity. The SGO/Fe_3_O_4_-4 shown in Figure 5c manifests in a high-rate, ultra-long cycle life, and its charge capacity is sustained at 510 mAh g^−1^ up to 1000 cycles. Based on the above results, the SGO/Fe_3_O_4_-4 composite is the optimal product, which exhibits the best lithium storage performance.

Electrochemical impedance spectra (EIS) were also studied to investigate the excellent performance of SGO/Fe_3_O_4_-4 from another viewpoint. Figure 6 shows the Nyquist plots of three samples for fresh electrodes and cycled electrodes. All of the plots for fresh electrodes consist of the depressed semicircle in the high frequency regions connected to a sloping line in the low frequency regions (Figure 6a). It can be clearly observed that the SGO/Fe_3_O_4_-4 shows smaller interface resistances (R_sf_) and charge-transfer resistances (R_ct_) (98.12 Ω) in comparison with that of SG (404.46 Ω) and SGO (158.99 Ω) electrodes (Appendix A). The small resistances of SGO/Fe_3_O_4_-4 composite are mainly attributed to the hollow Fe_3_O_4_ nanostructures, which provide convenient access for the electrolyte to wet the electrode surface and also offer an additional transport channel for Li^+^ diffusion [27]. For the cycled electrodes (Figure 6b), two depressed semicircles are ascribed to the interface resistances (R_sf_) and charge-transfer resistances (R_ct_), which decrease significantly in contrast with those of the fresh electrodes (54.82 Ω for SG electrode, 68.05 Ω for SGO electrode, and 39.02 Ω for SGO/Fe_3_O_4_-4 electrode), benefiting the diffusion kinetics upon cycling. The small resistance for the cycled SGO/Fe_3_O_4_-4 electrode reduces the energy barrier of Li^+^ intercalation into graphite and benefits fast Li^+^ diffusion and charge transfer [13]. 

To further confirm the potential application of the SGO/Fe_3_O_4_ composites in commercial batteries, we used the SGO/Fe_3_O_4_-4 composite as the anode and LiCoO_2_ as the cathode to assemble a full cell (labeled as SGO/Fe_3_O_4_-4/LiCoO_2_), and the corresponding electrochemical performance is discussed. The SGO anode was also assembled into a full cell with a LiCoO_2_ cathode (labeled as SGO/LiCoO_2_) for comparison. The related galvanostatic lithiation/delithiation curves of the SGO/Fe_3_O_4_-4/LiCoO_2_ full cell are exhibited in Figure 7a. The initial discharge and charge specific capacities for the SGO/Fe_3_O_4_-4/LiCoO_2_ full cell are 607 and 371 mAh g^−1^, respectively, with a low CE of 61.1%. Fortunately, the CE increases to 89% for the second cycle. As shown in Figure 7b, the SGO/Fe_3_O_4_-4/LiCoO_2_ full cell exhibits a higher specific capacity of 137 mAh g^−1^ after 100 cycles, which is higher than that of the SGO/LiCoO_2_ full cell (only 39 mAh g^−1^). When the full cell was cycled at 0.5 A g^−1^, the SGO/Fe_3_O_4_-4/LiCoO_2_ still maintained a reversible capacity of 79 mAh g^−1^ after 200 cycles (Figure 7c), which is better than SGO/LiCoO_2_. The rate capability of the SGO/Fe_3_O_4_-4/LiCoO_2_ full cell shows that the average specific capacities from 0.1 A g^−1^ to 5 A g^−1^ are 270 mAh g^−1^, 206 mAh g^−1^, 164 mAh g^−1^, 128 mAh g^−1^, 94 mAh g^−1^, and 54 mAh g^−1^, respectively (Figure 7d). An average reversible capacity of 170 mAh g^−1^ can be maintained when the current density returns to 0.1 A g^−1^. Based on the above results, the SGO/Fe_3_O_4_-4/LiCoO_2_ full cell delivers a higher specific capacity and better rate performance than SGO/LiCoO_2_. Surprisingly, the button full cell can easily provide sufficient power to light the light-emitting diode (LED) (Figure 7e), which lasts about 40 min. In our case, the capacity decay for the SGO/Fe_3_O_4_-4/LiCoO_2_ full cell can be found upon cycling. The reasons can be summarized as follows. First, the formed SEI layer on the surface of SGO/Fe_3_O_4_ is unstable from the three cycles of electrochemical activation. Second, the large volume changes of Fe_3_O_4_ during repeated cycles destroy the unstable SEI film, thus, regeneration and overgrowth of SEI film will consume more Li^+^ [41], resulting in the rapid capacity decay. Third, the electrode pulverization and insufficient electrolytes may be another reason [42]. This result implies that the Fe_3_O_4_ modified spherical-graphite composite is able to replace the commercial graphite for LIBs.

## 4. Conclusions

In summary, the Fe_3_O_4_ hollow nanosphere-modified SGO composites (SGO/Fe_3_O_4_) have been successfully prepared by the initial oxidation treatment of graphite and subsequent solvothermal synthesis. Among all of the composites, the SGO/Fe_3_O_4_-4 for half-cell exhibits the best lithium storage performance. Its charge capacity can reach as high as 806 mAh g^−1^ after 200 cycles at 0.5 A g^−1^, which is far higher than that of SG and SGO. When the electrode was cycled at 2 A g^−1^, the composite achieves a charge capacity of 510 mAh g^−1^ over 1000 cycles. The superior high-rate lithium storage performance is mainly attributed to its specially designed micro-nanostructure. Besides the half-cell, the SGO/Fe_3_O_4_-4/LiCoO_2_ full cell has been investigated, which exhibits higher capacity, better rate capacity, and better cycling performance than the SGO/LiCoO_2_ full cell. The low-cost synthesis method and eminent electrochemical performance of SGO/Fe_3_O_4_ composites demonstrate its promising application as a replacement for current graphite LIBs.

## Figures and Tables

**Figure 1 nanomaterials-09-00996-f001:**
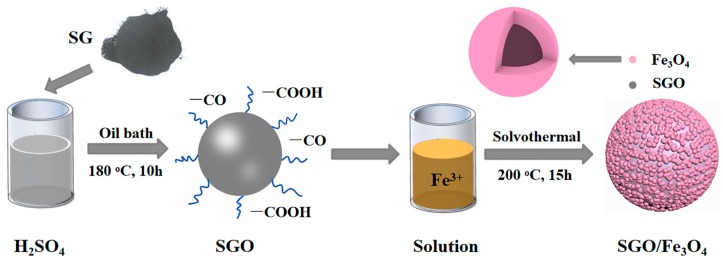
Preparation process of SGO/Fe_3_O_4_ composites.

**Figure 2 nanomaterials-09-00996-f002:**
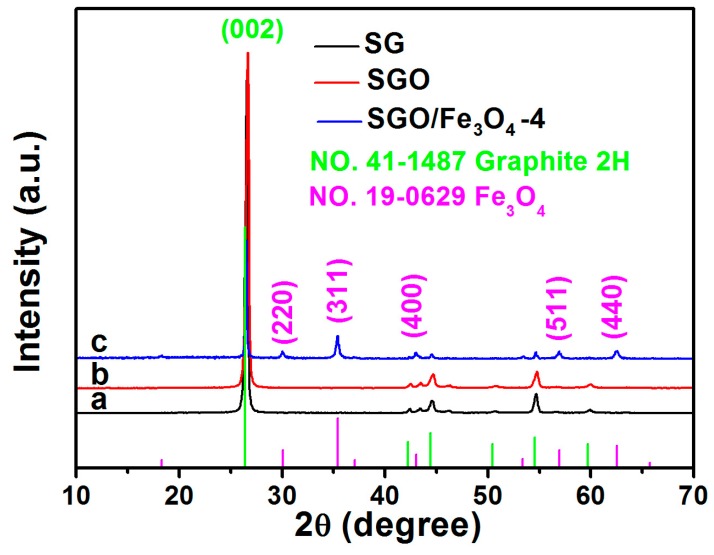
XRD patterns of (**a**) SG, (**b**) SGO, and (**c**) SGO/Fe_3_O_4_-4 composite.

**Figure 3 nanomaterials-09-00996-f003:**
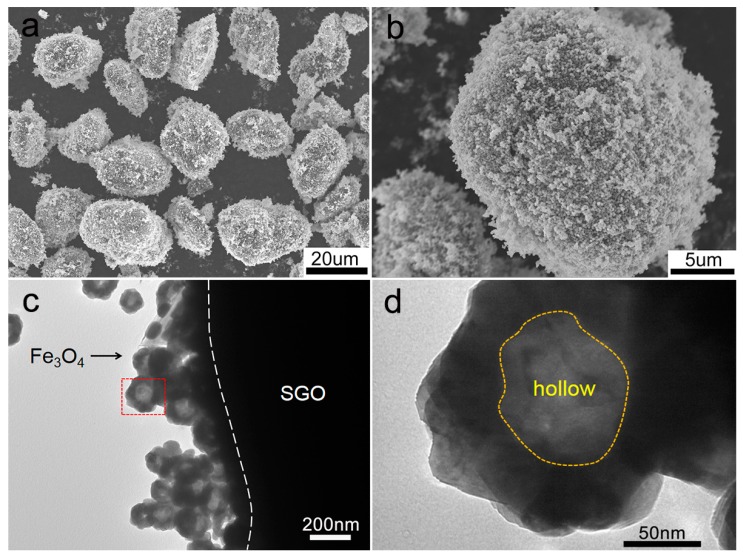
(**a**,**b**) SEM images and (**c**,**d**) TEM images of SGO/Fe_3_O_4_-4.

**Figure 4 nanomaterials-09-00996-f004:**
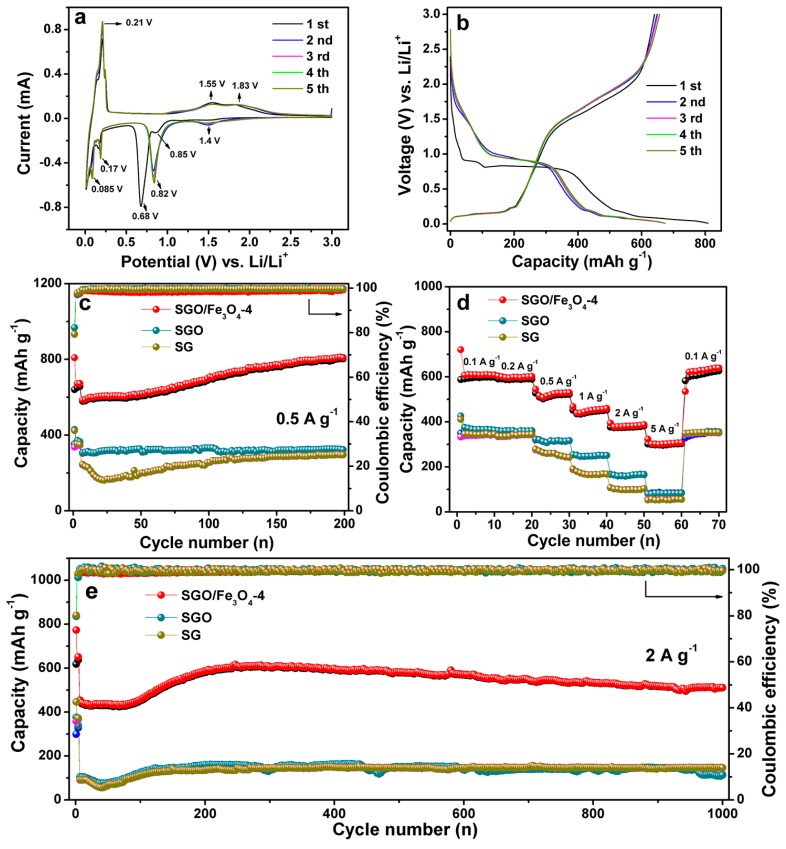
(**a**) CV curves of SGO/Fe_3_O_4_-4 at a scanning rate of 0.1 mV s^−1^. (**b**) Galvanostatic lithiation/delithiation curves of SGO/Fe_3_O_4_-4 at 0.1 A g^−1^. (**c**) Cycling performance at 0.5 A g^−1^ for 200 cycles, (**d**) rate capability, and (**e**) cycling performance of three samples at 2 A g^−1^ for 1000 cycles.

**Figure 5 nanomaterials-09-00996-f005:**
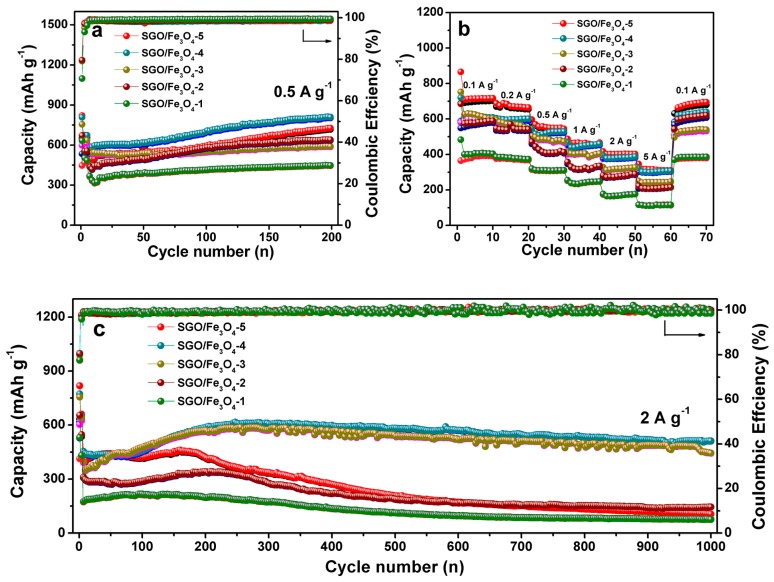
(**a**) Cycling performance at 0.5 A g^−1^ for 200 cycles, (**b**) rate capability, and (**c**) long-periodic cycling performance of different composites at 2 A g^−1^ for 1000 cycles.

**Figure 6 nanomaterials-09-00996-f006:**
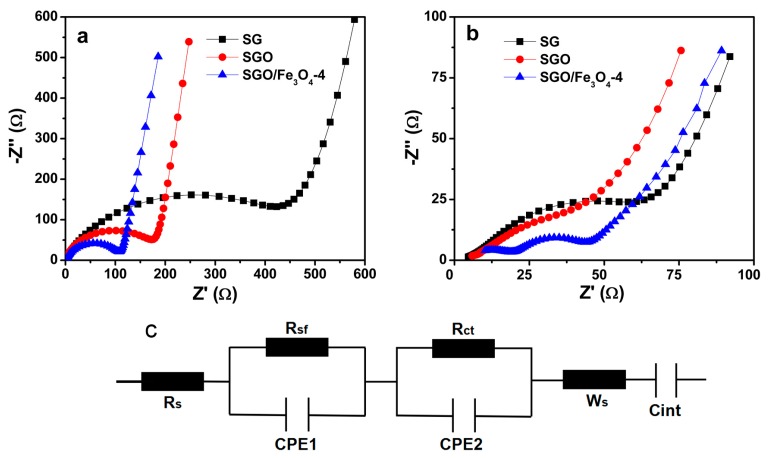
Nyquist plots of three samples: (**a**) fresh electrodes, (**b**) cycled electrodes, and (**c**) equivalent circuit diagram for Nyquist plots.

**Figure 7 nanomaterials-09-00996-f007:**
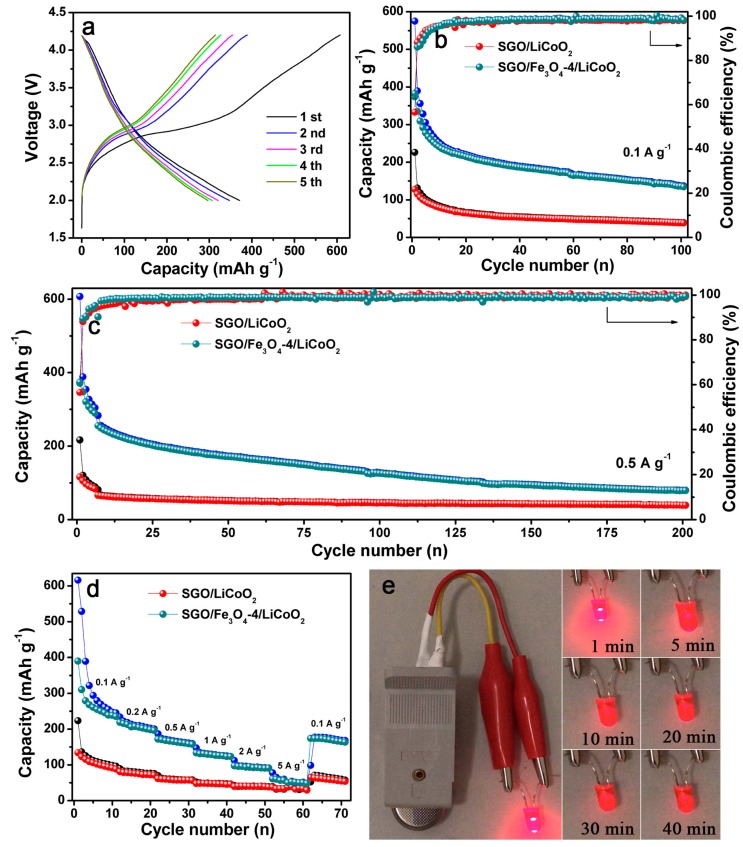
(**a**) Galvanostatic discharge/charge curves of the SGO/Fe_3_O_4_-4/LiCoO_2_ full cell between 2 V and 4.2 V at 0.1 A g^−1^, (**b**,**c**) cycling performance comparison of the SGO/LiCoO_2_ and SGO/Fe_3_O_4_-4/LiCoO_2_ full cells at 0.1 A g^−1^ and 0.5 A g^−1^, respectively, (**d**) rate capability and (**e**) picture of light-emitting diode (LED) powered by the assembled SGO/Fe_3_O_4_-4/LiCoO_2_ full cell.

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
