# Peer review of "Fe3O4 Hollow Nanosphere-Coated Spherical-Graphite Composites: A High-Rate Capacity and Ultra-Long Cycle Life Anode Material for Lithium Ion Batteries"

_nanomaterials, 2019, doi:10.3390/nano9070996_

Round 1
Reviewer 1 Report
Overall, this is a clear, concise, and well-written manuscript.
Author Response
Thank you very much for your reply on June 05, 2019, regarding our manuscript entitled "Fe3O4 hollow nanospheres modified spherical-graphite composites as high-rate capacity and ultra-long cycle life anode materials for lithium ion batteries"
Reviewer 2 Report
The title is confusing, please consider rewording as it took several times reading through to make sense.
The grammar needs to be improved. Although understandable, numerous errors are present throughout the text.
Page 1. Line 34. "Besides, because..." This sentence seems to be making the point that graphite is known to exfoliate in certain electrolytes. I don't see that as relevant here as electrolytes and SEI's are tailored to deal with these issues.
Page 1. Line 36. please add something about what conditions lead to lithium dendrite formation.
Page 1, Line 41. "The S-doped graphite...." This sentence is a rather abrupt change from the previous sentence. You need a transition sentence or introductory phrase to the sentence.
Page 2, Line 47. I would not agree with the statement "As we all know....." You could say "It has been shown that...." and then give references.
Page 2, Line 52. Please define the abbreviation SG.
Page 2, Line 71. What alcohol are you using?
Page 4, Line 131, Please show additional TEM to support your conclusion that the nano spheres are hollow.
Figure 3. It is difficult to see the 5 um bar in 3b.
Page 5, Line 148. Supplemental figure S3c missing.
Page 5, Line 155. "The capacity increase is ..." Please be more specific as to what pseudo-capacitive behavior and activation reactions are.
Page 8, Line 215 "...is possibly due to the generation of mesoporous..." is speculative. This should be removed unless you can show some supporting information.
Author Response
1.Comments: The title is confusing, please consider rewording as it took several times reading through to make sense.
Response: Thank you for the suggestion. The title has been revised as “Fe3O4 hollow nanospheres coated spherical-graphite composites: a high-rate capacity and ultra-long cycle life anode material for lithium ion batteries”.
2.Comments: The grammar needs to be improved. Although understandable, numerous errors are present throughout the text.
Response: Thank you for the reminder. We have improved the grammar of manuscript carefully, and some grammatical mistakes have been corrected. The revision has been highlighted by yellow.
3.Comments: Page 1. Line 34. "Besides, because..." This sentence seems to be making the point that graphite is known to exfoliate in certain electrolytes. I don't see that as relevant here as electrolytes and SEI's are tailored to deal with these issues.
Response: Thank you for the advice. The corresponding statement has been removed in the text.
4.Comments: Page 1. Line 36. Please add something about what conditions lead to lithium dendrite formation.
Response: Thank you for the suggestion. Some sentence has been added into the text, the details are as follows: “the voltage plateau of graphite is low, close to the lithium, which will result in the growth of lithium dendrites on the graphite surface. The formed lithium dendrites can not only reduce the battery capacity, cause irreversible consumption of Li and increase charge-transfer resistance, but also lead to serious safety accidents [6].”.
5.Comments: Page 1, Line 41. "The S-doped graphite...." This sentence is a rather abrupt change from the previous sentence. You need a transition sentence or introductory phrase to the sentence.
Response: Thank you for the suggestion. The related transition sentence has been added into the text, which has been highlighted by yellow. The details are as follows: “Besides, to hybrid with a high capacity anode material can improve the lithium storage performance of graphite. For example, the S-doped graphite....”.
6.Comments: Page 2, Line 47. I would not agree with the statement "As we all know....." You could say "It has been shown that...." and then give references.
Response: Thank you for the suggestion. The phrase “As we all know...” has been revised as “It has been shown that....” in the text.
7.Comments: Page 2, Line 52. Please define the abbreviation SG.
Response: Thank you for the reminder. The abbreviation SG has been defined in the text.
8.Comments: Page 2, Line 71. What alcohol are you using?
Response: Thank you for the question. The alcohol used in our experiment is absolute ethyl alcohol, its purity is 99.7 wt%. The related revision has been highlighted by yellow.
9.Comments: Page 4, Line 131, please show additional TEM to support your conclusion that the nanospheres are hollow. Figure 3. It is difficult to see the 5 um bar in 3b.
Response: Thank you for the advice. The additional magnified TEM image has been added into Figure 3 to confirm the Fe3O4 nanospheres are hollow. Besides, the colour of bar has been revised. The details are as follows:
Figure 3. (a) and (b) SEM images, (c) and (d) TEM images of SGO/Fe3O4-4.
10.Comments: Page 5, Line 148. Supplemental figure S3c missing.
Response: Thank you for the question. Figure S3 shows the galvanostatic lithiation/delithiation curves of (a) SG and (b) SGO at 0.1 A g-1. There are only two figures, and Figure S3c is inexistent.
11.Comments: Page 5, Line 155. "The capacity increase is ..." Please be more specific as to what pseudo-capacitive behavior and activation reactions are.
Response: Thank you for the good question. According to your advice, we have explained the reason of capacity increase in detail, and corresponding revision has been highlighted by yellow. The details are as follows: The capacity increase may be attributed to the following reasons: the first reason is the self-activation process of Fe3O4 active materials upon repeated cycles [33-35]. The particle size of Fe3O4 will decrease. The increased specific surface area due to the small size will form more active sites, and improve the electrolyte accessibility, which can provide more surface-related capacitance [33-35]. Second, as the cycle number increases, a more stable SEI film will form on the surface of electrode material, which is beneficial for the lithium ion storage [36]. Third, the reversible growth of a polymeric gel-like film resulting from kinetically electrolyte degradation can also lead to the capacity increase [37-39].
12.Comments: Page 8, Line 215 "...is possibly due to the generation of mesoporous..." is speculative. This should be removed unless you can show some supporting information.
Response: Thank you for the suggestion. The related sentence has been remove
Reviewer 3 Report
Review of the manuscript entitled ‘Fe3O4 hollow nanospheres modified spherical-graphite composites as high-rate capacity and ultra-long cycle life anode materials for lithium ion batteries’.
The manuscript is about the electrochemical performance of Fe3O4/graphite composite. It showed its superior performance to spherical graphite alone.
1. Page 5: the lines 162 and 163 are not clear. There are many values of capacities. It should explain their corresponding current densities.
2. The performance of SGO/ Fe3O4 is high compared to SGO or SG alone. However, the explanation of such high performance of SGO/ Fe3O4 (line 168-174 in page 5) is not convincing. The reviewer suggests that the authors test the performance of Fe3O4 without SGO or SG (thus, the active material will be only Fe3O4) in order to support the synergetic effect of the combination of Fe3O4 and SGO.
3. The authors should mention how the mass of the active material in an electrode is calculated for SGO/ Fe3O4. In addition, the authors described the loading of material is about 1 mg. The reviewer found this value of loading is too rough considering that the mass variation of Fe3O4 is from 11. 6 to 48 % of Fe3O4 in the composite. The reviewer suggests that the authors should indicate more precisely the mass of the active material and the thickness of the electrode for each sample from SGO/ Fe3O4-1 to SGO/ Fe3O4-5 as well as SGO and SG.
4. The authors should indicate the mass loading and the thickness of LiCoO2 electrode.
5. About a LED light part, there are various types of LED requiring different power and current. It is not so surprising one cell can light up a LED. The reviewer suggests that the authors indicate what type of LED (power, current etc.) and the reviewer wonders how long the LED was running with a single time charged cell.
6. In Fig. 7b, why does the capacity of a full cell SGO/ Fe3O4/LiCoO2 fade rapidly compared to a SGO/LiCoO2 full cell ?
7. The rests are following
· English needs to be corrected and polished.
· The numbers in ‘Fe3O4’ should be subscript.
· The TEM image of Fig.3 (d) is elongated. It needs to show the image ‘as-it-is’. (probably a mistake of converting a file)
· As the unit of ‘V’ is relative, it should be addressed ‘versus Li+/Li ‘.
· The authors used abbreviations such as Fe2O3-G, SG and SGO at the beginning without the full description.
· Is the unit of loading weight, mg cm-1 correct ?
Author Response
1.Comments: Page 5: the lines 162 and 163 are not clear. There are many values of capacities. It should explain their corresponding current densities.
Response: Thank you for the question. The corresponding current densities have been explained in the text. The details are as follows: “The average charge capacity cycled at 0.1, 0.2, 0.5, 1, 2 and 5 A g-1 is 595, 589, 516, 444, 376 and 299 mAh g-1, respectively.”
2.Comments: The performance of SGO/Fe3O4 is high compared to SGO or SG alone. However, the explanation of such high performance of SGO/Fe3O4 (line 168-174 in page 5) is not convincing. The reviewer suggests that the authors test the performance of Fe3O4 without SGO or SG (thus, the active material will be only Fe3O4) in order to support the synergetic effect of the combination of Fe3O4 and SGO.
Response: Thank you for the good question. According to your advice, the lithium storage performance of Fe3O4 hollow nanospheres has been tested, which have been added into Figure 4 for comparison. The data can be seen as follows:
Figure 4. (a) CV curves of SGO/Fe3O4-4 at a scanning rate of 0.1 mV s-1. (b) Galvanostatic lithiation/delithiation curves of SGO/Fe3O4-4 at 0.1 A g-1. (c) Cycling performance at 0.5 A g-1 for 200 cycles, (d) rate capability and (e) cycling performance of four samples at 2 A g-1 for 1000 cycles.
It can be observed that Fe3O4 hollow nanospheres show higher specific capacity and better rate performance than that of SGO/Fe3O4-4. The cycling performance of SGO/Fe3O4-4 is comparative to that of hollow Fe3O4 nanospheres. Thus, the synergetic effect of the combination of Fe3O4 and SGO is not obvious. The Fe3O4 nanospheres are just acted as surface modifier to improve the lithium storage performance of SGO. The phrase “synergetic effect” has been deleted on the text, and related revision has been highlighted by yellow. Besides, the electrochemical performance of Fe3O4 hollow nanospheres has been added into supporting information (Figure S5).
3.Comments: The authors should mention how the mass of the active material in an electrode is calculated for SGO/Fe3O4. In addition, the authors described the loading of material is about 1 mg. The reviewer found this value of loading is too rough considering that the mass variation of Fe3O4 is from 11.6 to 48 % of Fe3O4 in the composite. The reviewer suggests that the authors should indicate more precisely the mass of the active material and the thickness of the electrode for each sample from SGO/Fe3O4-1 to SGO/Fe3O4-5 as well as SGO, SG and LiCoO2 electrode.
Response: Thank you for the good question. The loading weights (about 1 mg cm-2) of active materials we mentioned in the text are the weights of overall composites including SGO and Fe3O4. The weight of SGO or Fe3O4 in the composite can be calculated according to TG results. Besides, the thickness of the electrode for each sample from SGO/Fe3O4-1 to SGO/Fe3O4-5 as well as SGO, SG and LiCoO2 has been estimated respectively, the corresponding SEM images are shown as follows. The corresponding areal mass of the active material and the thickness of the electrode for each sample from SGO/Fe3O4-1 to SGO/ Fe3O4-5 as well as SGO, SG and LiCoO2 electrodes have been marked on Figure S1.
Figure S1. The areal loading weights and thickness of (a) SG, (b) SGO, (c) SGO/Fe3O4-1, (d) SGO/Fe3O4-2, (e) SGO/Fe3O4-3, (f) SGO/Fe3O4-4, (g) SGO/Fe3O4-5, (h) LiCoO2-1 (matching with SGO) and (i) LiCoO2-2 (matching with SGO/Fe3O4-4), respectively.
4.Comments: About a LED light part, there are various types of LED requiring different power and current. It is not so surprising one cell can light up a LED. The reviewer suggests that the authors indicate what type of LED (power, current etc.) and the reviewer wonders how long the LED was running with a single time charged cell.
Response: Thank you for the suggestion. The voltage of LED we used is 1.8-2 V, and current is 15 mA. According to the video, the LED can run about 40 min with a single
time charged cell. The corresponding picture has been added into Figure 7. The electrochemical performance of SGO/Fe3O4-4/LiCoO2 full cell is not very ideal, the related work is under way.
Figure 7e. The picture of light-emitting diode (LED) powered by the assembled SGO/Fe3O4-4/LiCoO2 full cell.
5.Comments: In Fig. 7b, why does the capacity of a full cell SGO/ Fe3O4/LiCoO2 fade rapidly compared to a SGO/LiCoO2 full cell?
Response: Thank you for the good question. The reason of capacity decay for full cell can be attributed to three points: first, the loss of active Li [1]; second, the loss of active materials [2]; third, the internal polarization; in our case, the first reason is the main factor to cause the capacity loss. Compared with spherical-graphite, the formed SEI layer on the surface of SGO/Fe3O4 is not stable just by the electrochemical activation for three cycles. Besides, the large volume changes of Fe3O4 upon cycling will destroy the unstable SEI film, thus, regeneration and overgrowth of SEI film will consume more Li+ [1], resulting in the rapid capacity decay. The electrode pulverization and insufficient electrolyte may be another reason [2].
References:
[1] O. Vargas, A. Caballero, J. Morales, E.R. Castellon, Contribution to the understanding of capacity fading in graphene nanosheets acting as an anode in full Li-ion batteries. ACS Appl. Mater. Interfaces 2014, 6, 3290−3298.
[2] P. Xia, H. Lin, W. Tu, X. Chen, X. Cai, X. Zheng, M. Xu, W. Li, A novel fabrication for manganese monoxide/reduced graphene oxide nanocomposite as high performance anode of lithium ion battery. Electrochimica Acta 2016, 198, 66-76.
The details in the text are as follows: “Anyway, in our case, the capacity decay for SGO/Fe3O4-4/LiCoO2 full cell can be found upon cycling. The reasons can be summarized as follows: First, the formed SEI layer on the surface of SGO/Fe3O4 is not stable just by the electrochemical activation for three cycles. Second, the large volume changes of Fe3O4 during repeated cycles will destroy the unstable SEI film, thus, regeneration and overgrowth of SEI film will consume more Li+ [41], resulting in the rapid capacity decay. Third, the electrode pulverization and insufficient electrolyte may be another reason [42].”
6.Comments: The rests are following:
· English needs to be corrected and polished.
· The numbers in‘Fe3O4’should be subscript.
· The TEM image of Fig. 3 (d) is elongated. It needs to show the image‘as-it-is’. (probably a mistake of converting a file)
· As the unit of‘V is relative, it should be addressed‘versus Li+/Li‘.
· The authors used abbreviations such as Fe2O3-G, SG and SGO at the beginning without the full description.
· Is the unit of loading weight, mg cm-1 correct ?
Response: Thank you for the good suggestion. According to the reminder, the English has been corrected. The numbers in “Fe3O4” are subscript. The TEM image of Fig. 3 (d) has been adjusted. The abbreviations such as Fe2O3-G, SG and SGO at the beginning have been defined. The unit of‘V’in Figure 4a and 4b has been revised as V vs Li+/Li”. The unit of loading weight “mg cm-1” has been revised as “mg cm-2”. Besides, the other mistakes have been corrected, and all the related revision has been highlighted by yellow in the text.
Round 2
Reviewer 3 Report
There are a few points unclear and it requires more characterizations.
The yellow lines (160-165) are not understandable. What is ‘self-activation process’ of Fe3O4 ? Why does SEI layer become more stable upon cycling? And the third reason is not understandable. Therefore, the effect of Fe3O4 hollow structure is not clear. The reviewer strongly suggests that Fe3O4 alone should be tested electrochemically.
The reviewer suggests following characterizations:
1) FT-IR of graphite before treatment, functionalized graphite, and the composite of SGO/Fe3O4
2) Porosity of Fe3O4 and specific surface area of graphite and Fe3O4
About the capacity decay of the SGO/Fe3O4/LiCoO2 full cell, the authors explained that the degradation reasons can be due to the unstable SEI layer on the anode, the large volume change of Fe3O4 and pulverization. However, these issues should have appeared during the cycling of a half-cell. The capacity retention of a SGO/Fe3O4 half-cell was about 65 % while that of a full-cell was only about 17%. The authors should do a cycling test of LiCoO2 half-cell because LiCoO2 cathode may provide poor cyclability.